# Iron Overload in Brain: Transport Mismatches, Microbleeding Events, and How Nanochelating Therapies May Counteract Their Effects

**DOI:** 10.3390/ijms25042337

**Published:** 2024-02-16

**Authors:** Eleonora Ficiarà, Ilaria Stura, Annamaria Vernone, Francesca Silvagno, Roberta Cavalli, Caterina Guiot

**Affiliations:** 1School of Pharmacy, University of Camerino, 62032 Camerino, MC, Italy; eleonora.ficiara@unicam.it; 2Department of Neurosciences, Università degli Studi di Torino, 10125 Torino, TO, Italy; annamaria.vernone@unito.it (A.V.); caterina.guiot@unito.it (C.G.); 3Department of Oncology, Università degli Studi di Torino, 10126 Torino, TO, Italy; francesca.silvagno@unito.it; 4Department of Drug Science and Technology, Università degli Studi di Torino, 10125 Torino, TO, Italy; roberta.cavalli@unito.it

**Keywords:** iron, Alzheimer’s disease, neurodegeneration, chelation

## Abstract

Iron overload in many brain regions is a common feature of aging and most neurodegenerative diseases. In this review, the causes, mechanisms, mathematical models, and possible therapies are summarized. Indeed, physiological and pathological conditions can be investigated using compartmental models mimicking iron trafficking across the blood–brain barrier and the Cerebrospinal Fluid-Brain exchange membranes located in the choroid plexus. In silico models can investigate the alteration of iron homeostasis and simulate iron concentration in the brain environment, as well as the effects of intracerebral iron chelation, determining potential doses and timing to recover the physiological state. Novel formulations of non-toxic nanovectors with chelating capacity are already tested in organotypic brain models and could be available to move from in silico to in vivo experiments.

## 1. Introduction

Dementia, and in particular the most diffused sporadic or non-familial Alzheimer’s Disease (AD) subtype, is an emergent ‘social’ pandemic whose main hallmarks are cognitive and memory impairment, neuronal and synaptic loss, hippocampus shrinking, amyloid β plaques and cerebral amyloid angiopathy, neurofibrillary tangles and glial responses.

Older age is the most important known risk factor for sporadic AD, but the complex etiology remains under continuous investigation. It is a multifactorial pathology, linking population risk factors, e.g., nanopollution [1], sleep disruption [2], or even bacterial infections [3,4].

There is an unmet need for specific, cost-effective, and easily detectable biomarkers for (elder) population screening to intercept intermediate phases of pathological evolution, from amyloid β plaques to astrocyte reactivity (Glial Fibrillary Acidic Protein—GFAP), Tau tangle formation and severe cognitive impairment.

AD diagnostics still show the main criticalities. First, biomarker detection in cerebrospinal fluid (CSF) is very invasive, and there is an urge to use other biological fluids (e.g., tears, saliva, sweat) in addition to serum. Moreover, although PET imaging detects clusters of amyloids (plaques) or tau (neuro-fibrillary tangles) proteins, it is a very expensive and not very common resource compared with MRI. The potentialities of the latter technique for early detection of hippocampal deterioration [5,6] and indirect evaluation of plaques/tangles by colocalization of magnetic elements (such as iron) are not fully exploited. Finally, cognitive examinations and neuropsychological tests are severely biased by the patient’s cultural level, language, and social/environmental impairments [7].

In this review, the role of iron overload in AD pathogenesis, and in particular the mechanisms of transport and accumulation, are summarized. Moreover, some mathematical models are reported, focusing on possible therapies targeting iron deposits and their link with AD and/or dementia.

## 2. Iron: A (New) Responsive Biomarker

Iron is an essential metal for many physiological functions, and it is mainly transported by proteins that ‘isolate’ the iron ions from the cells. Its concentration varies due to many reasons, such as age, ferroptosis, microbleeds, and other alterations of transport mechanisms. Moreover, iron concentration changes depending on the brain area [8].

Notwithstanding its fundamental role, iron accumulation can also contribute to central nervous system (CNS) disorders, such as AD and Parkinson’s disease (PD). Indeed, iron is potentially very toxic, and its involvement in AD progression has been deeply investigated [9,10,11,12].

Iron accumulation could also depend on genetic and environmental factors.

It has been hypothesized that, following exposure to environmental pollutants, the perturbation of functional iron homeostasis may be the mechanism leading to adverse biological effects [13].

To date, mutations in 13 genes have been associated with autosomal dominant (FTL), autosomal recessive (ATP13A2, PANK2, PLA2G6, FA2H, CP, C19orf12, COASY, GTPBP2, DCAF17, VAC14), and X-linked forms of neurodegeneration with brain iron accumulation (NBIA) (WDR45, RAB39B) and are involved in a wide range of molecular processes affecting mitochondrial function, coenzyme A metabolism, lipid metabolism and autophagy [14]. Additionally, genetic alterations in iron uptake or homeostasis could dramatically influence responses to environmental toxins, given the well-known role of iron in oxidative stress [15].

### 2.1. Iron Detection

Iron is currently dosed in plasma to detect systemic pathologies; its commonly used indicators are hemoglobin, ferritin, transferrin, and transferrin saturation.

Standard and non-standard iron indicators, both in serum and in CSF, and relative alterations detectable by the indicators of iron status are reviewed in [16].

Moreover, iron in the CNS can be quantified using direct or indirect methods.

It can be directly measured in the CSF with non-clinical standard techniques, e.g., Graphite Furnace Atomic Absorption Spectrometry (GF-AAS) [16,17]. Furthermore, marketed kits are still unreliable due to the low concentration and different biological forms of iron in the CNS, requiring very sensitive and reliable methods for the different iron forms.

Indirect methods based on proper MRI sequences are still under investigation. It has been shown that some quantitative MRI methods, such as transverse relaxation rate R2* (1/T2*) and quantitative susceptibility mapping (QSM), are sensitive to brain iron, based on their strong correlation with post-mortem iron measures and highly variable concentrations in specific anatomical structures [18,19]. The relaxation time constant (1/T2* = 1/T2 + 1/T2′) is sensitive to local inhomogeneities in the brain tissue, also due to iron deposits. However, R2* requires multi-echo scans that are prone to subject motion and other errors [20], while QSM also accounts for other pathophysiological susceptibility changes, although, especially in the basal ganglia and other grey matter structures, it is mainly determined by ferritin iron saturation [21,22].

The extensive use of MRI in examining morphological properties is facilitated by automatic computational algorithms (e.g., Freesurfer and FSL), which draw insights from structural MR images [23].

Perturbed iron neurochemistry and the associated formation of aberrantly aggregated proteins have been implicated in the development of several degenerative brain disorders, including AD. PET is a broadly employed technique for investigating neurodegenerative diseases. Intrinsic links exist between these diseases and protein aggregates, including Tau or amyloid β (Aβ) proteins, visible through diverse PET radiotracers.

Aβ plaques, identifiable through PET, can overlap with sites of iron deposition, which should locally be superposed to the same regions evidenced by MR images, where the T2* shortening effect of paramagnetic iron highlights the contrast between plaques and surrounding tissues. Additionally, MRI images, with their greater accessibility, offer specific sequences designed to be sensitive to the magnetic properties of small iron deposits. A comparative analysis between PET and MRI can validate this hypothesis, potentially guiding a more effective implementation of MRI examinations. Furthermore, morphological distinctions in brain regions, such as hippocampal shrinking and abnormal cortical features discernible through MRI, provide valuable insights that can be distinguished between control subjects and AD patients.

Moreover, nanoscale resolution soft X-ray spectromicroscopy is used to examine the interaction of Aβ, fundamentally implicated in the development of AD, and ferric (Fe^3+^) iron [24]. The co-aggregation of Aβ and iron is known to affect iron chemistry, resulting in the chemical reduction of Fe^3+^ into reactive and potentially toxic ferrous (Fe^2+^) and zero-oxidation (Fe-0) states. Combining nanoscale-resolution imaging and high chemical sensitivity, nanoscale (i.e., sub-micron) variations in both the iron oxidation state and the organic composition of Aβ were observed, supporting the hypothesis that Aβ is responsible for altering iron neurochemistry and that this altered chemistry is a factor in neurodegenerative processes documented in AD [24].

### 2.2. Iron-Related Damaging Mechanisms in CNS

There are many damaging mechanisms triggered by iron accumulation, briefly summarized in Table 1. In this section, they are discussed and analyzed.

#### 2.2.1. Direct Mechanisms

It is well known that iron excess is directly responsible for oxidative stress and inflammation by the Fenton and Haber–Weiss reactions (A1 in Table 1), which occur in the presence of iron and produce Reactive Oxygen Species (ROS), specifically OH (hydroxyl radicals) from H_2_O_2_ (hydrogen peroxide) and superoxide (O^2−^). Such oxidative stress targets the brain metabolism, i.e., inducing the so-called ‘ferroptosis’ [25]. Ferroptosis is an iron-dependent cell death characterized by the accumulation of lipid ROS, which is morphologically, biochemically, and genetically different from other forms of cell death [26]. The ferroptosis pathway can be initiated through transferrin (Tf) endocytosis linked to Tf receptor 1 (TfR1). Ferric iron (also stored in ferritin) is released from the TfR1 complex, reduced to ferrous iron, and can be stored in ferritin or remain in the cytoplasm (as a labile iron pool). It is able to generate ROS through the Fenton reaction and lipid peroxidation.

Moreover, iron may induce the formation of plaques (Aβ, Tau) (A2 in Table 1), and early detection may therefore reduce or control such aggregation, which is responsible for the macroscopic symptomatology. Indeed, amyloid plaques have been described as “metallic sinks” because remarkably high concentrations of Cu^2+^ and Fe^3+^ have been found within these deposits in AD [27]. This dangerous liaison has been investigated for a long time, showing a slower stabilization of the more damaging ferrous (Fe^2+^) to Fe^3+^ in physiological conditions in the presence of aluminum salts as well as in the presence of Aβ fragments [28]. More recently, early plaque formation in the cortex and hippocampus in a APP/PS1 mouse model was related to elevated iron content [29] and was associated with hemoglobin (Hb) binding to Aβ localized within the plaque and vascular amyloid deposits in post-mortem AD brains [30]; namely, the formation of an envelope-like structure composed of Aβ surrounding the Hb droplets was observed in APP/PS1 transgenic mice [30]. Interestingly, investigations on the heme–Aβ complexes were performed, along with the biological model for aggregated amyloid on conducting surfaces, developed as a powerful tool to investigate the reactivity of redox-active metal/cofactor bound to fibrillar and oligomeric Aβ and to screen drugs designed to target these [31]. MRI QSM sequences were also used to longitudinally monitor Aβ accumulation occurring concomitantly to iron deposition in vivo [32], and Tau aggregation related to cortical iron accumulation was shown using MRI and Tau-PET [33]. Furthermore, increased glutamate and glutamine levels in the CSF of AD patients may be due to the build-up of Aβ oligomers in the AD brain [34].

Since the AD-related cognitive decline reflects an alteration in the astroglia-neuronal network, alterations of the bioelectric neuronal activity (A3 in Table 1) are expected. Other effects of iron overloading in the hippocampus of the AD APP/PS1 mice model recently investigated [35] include an impact on the proprotein convertase Furin, which catalyzes the proteolytic maturation of a large number of prohormones and proproteins, and on the maturation of the brain-derived neurotrophic factor (BDNF). Both of the above fundamental components were reduced or even suppressed by the effect of iron overload and partly recovered following iron chelation. Furthermore, progressive iron accumulation in the *substantia nigra* in the aged brain is a risk factor for PD and other neurodegenerative diseases. The adaptive cellular response of the *substantia nigra* dopaminergic neurons upon age-dependent iron accumulation was investigated [36]. The disruption of the interplay of dopamine, alpha-synuclein, and iron pathways may synergize to promote pathology and drive the unique vulnerability to disease states [37].

Finally, iron is strictly necessary for the proliferation of microorganisms (A4 in Table 1), such as bacteria, viruses, and fungi [38]. Nutritional immunity is another host defense mechanism that limits the availability of essential metals, such as iron, from invading bacterial pathogens, and iron starvation has been proposed as a potential antibacterial strategy [39]. The mechanism used by bacteria, e.g., Staphylococcus aureus, to extract iron from Hb has been recently identified [40] in two receptors, IsdB and IsdH, exposed to the surface of the bacterium, being the first of them formed by NEAT (Near Iron Transports) domains, which bind heme from the ferric form of Hb, accelerating its release up to 2000 fold.

#### 2.2.2. Indirect Mechanisms

As an indirect effect of A1, iron-induced oxidation (B1 in Table 1) has an impact on the pathways related to lipid metabolism in AD and PD, regulated by the Apolipoprotein, in particular ApoE, which is known to be one of the most validated risk markers for sporadic AD, with ApoE4 representing the strongest genetic risk factor for the development of late-onset AD [41]. It has been shown that ApoE acts as a potent inhibitor of ferroptosis by activating the PI3K/AKT pathway and inhibiting the autophagic degradation of ferritin (ferritinophagy) [42]. Moreover, ferritin levels in the CSF are increased in patients with the ApoE4 allele, and increased CSF ferritin levels are associated with earlier disease onset [43]. CSF ferritin can predict AD outcomes and is regulated by ApoE [44], having a role in the clinicopathological progression of AD [45].

As an indirect consequence of A2, cellular mitochondrial homeostasis can be influenced (B2 in Table 1). Aβ fibrils can aggregate in mitochondria and impair their function, leading to energy metabolism loss and elevated ROS production [46]. At least a portion of the Fe^2+^ acquired by neurons is expected to accumulate in mitochondria, where it is either stored in specialized structures (ferritin and frataxin) to prevent the generation of ROS or used by mitochondrial enzymes. The perturbation of mitochondrial iron homeostasis could deeply impact mitochondrial health and, consequently, neuronal function. Iron handling in hippocampal neurons showed an activity-dependent iron entry and mitochondria-mediated neurotoxicity [47]. The dysmetabolism of mitochondrial functional iron (enzyme-bound) is another indirect iron-related damaging mechanism (B2 in Table 1). To better understand the link between the function of mitochondrial proteins involved in iron homeostasis and the onset of AD, an investigation on the Uniprot [48] database was conducted. The used keywords were: “Alzheimer’s”, “iron”, “mitochondrial”, “dementia”, and “cognitive disorder”. These keywords were not only employed as simple search terms but were also incorporated as gene ontology (GO) terms to refine the results and ensure relevance to the results of the database exploration. After conducting cross-referencing analyses, the following proteins were identified as potential markers of AD associated with iron accumulation:Humanin belongs to a family of mitochondrial-derived peptides (MIDPs), and it is a peptide encoded by the mitochondrial genome. It is associated with life extension and possesses antioxidant and protective functions within the mitochondrion [49]. In AD, levels of Humanin decrease [50]. Notably, Humanin is produced in mitochondria and has the ability to enhance the survival of these organelles under conditions of excessive iron accumulation. Specific Single Nucleotide Polymorphisms (SNPs) of Humanin have been identified, with NP rs2854128 showing an inverse correlation with Humanin levels in the blood of elderly individuals [51]. This SNP may have implications for AD pathogenesis and could serve as a potential genetic marker.Appoptosin (Solute carrier family 25 member 38—SLC25A38), also known as Mitochondrial glycine transporter (GlyC) or SLC25A38, plays a crucial role in heme synthesis by facilitating the transport of glycine required for the initial step of heme biosynthesis into the mitochondrion. Appoptosin is involved in neuron death associated with neurodegeneration [52]. In AD, Appoptosin is upregulated [52]. The heme biosynthetic pathway involves the insertion of iron as the final step. Therefore, along with glycine, mitochondria must import iron using the transporter Mitoferrin (Mf) and store it in the Frataxin protein to prevent toxicity until it is utilized. An SNP (rs1768208, C/T) associated with progressive supranuclear palsy is located near the MOB gene on chromosome 3 and is correlated with increased expression of Appoptosin, whose gene is approximately 70 kb apart from MOB [53]. Moreover, in schizophrenia, a Genome-wide association analysis (GWAS) identified an association between SNP rs56055643 on chromosome 3, the expression of SLC25A38 (Appoptosin gene), and hippocampal dentate gyrus volume [54]. These associations indicate the potential involvement of Appoptosin in AD and other neurological disorders.Mitoferrin and Frataxin modulate the amount of iron imported into the mitochondrion and stored in a non-toxic form. They are regulated by hypoxia via HIF-1a [55]. In human degenerative pathologies, there is only one data point in Huntington Disease, in which both Mf2 increases and frataxin (the mitochondrial analogue of ferritin) decreases [56], while in human AD there is no experimental evidence such as protein expression.Phosphatidylinositol-binding clathrin assembly protein (PICALM) acts as a modulator of the internalization of TfR (transferrin receptor), therefore modulating the entry of iron into cells. The PICALM gene is considered a risk factor for LOAD (late-onset AD). The PICALM SNP rs10792832 has been studied in association with APOE4 and BIN1 SNPs in AD risk [57]. Furthermore, the SNP rs3851179 is included in the list of SNPs that correlate most with the risk of AD [58].APOE-ε4 is considered a strong risk factor for sporadic AD. As said before, it is an inhibitor of ferritinophagy [42]. The fluid concentrations of ApoE and its different isoforms in AD patients and among APOE genotypes remain controversial. In fact, one study reported that ApoE content in CSF is not an indicator of AD progression and that there is no association between plasma levels of total ApoE or its isoforms and AD biomarkers [59]. Instead, another study observed in the CSF of AD an increase in total ApoE content and an alteration of the ApoE protein, suggesting that function may be compromised [60]. Moreover, there is strong evidence for the association between AD and APOE-4 polymorphisms and for the association with at least two SNPs located less than 16 kb from APOE [61].

This interesting analysis reveals that several proteins linked to iron homeostasis are indeed modulated in AD and suggests that they could serve as biochemical markers for the disease. Further investigation is required to test their potential role.

Iron–calcium interplay (B3 in Table 1) can be investigated. The abnormal activity of the ryanodine receptors leading to Ca^2+^ dysregulation affects neuronal functionality in AD [62]. Indeed, calcium mediates synaptic plasticity and controls the expression of genes that are important during memorization [63]. Moreover, calcium accumulation in mitochondria can be a biomarker of the loss of iron homeostasis [64]. Calcium is also related to Aβ and cholesterol content in the cellular membrane [65]. The relationship, at the nanoscale level, between iron and calcium was deepened in recent studies [66] using X-ray spectromicroscopy.

Finally, several research groups have reported that gut dysbiosis is significantly associated with neuroinflammation, aggregation of Aβ, and an increase in oxidative stress during AD [67]. Then, the bacterial communities present in non-demented controls and AD subjects’ brains were profiled, and the results are consistent with a leaky blood–brain barrier (BBB) or lymphatic network that allows bacteria, viruses, fungi, or other pathogens to enter the brain [3] (B4 in Table 1). Recent investigations focused on how dysbiosis promotes and enhances oxidative stress, although the connection between specific bacteria and AD is not fully demonstrated [68]. An even more interesting study connects the peripheral Aβ concentration in the gut and in the brain in mice, investigating the possible mechanisms of migration and concluding that the bloodstream, more than transmission via the vagal nerve, is possibly involved [69].

### 2.3. Where Does the Iron Excess in the Brain Come From?

A very simple three-compartmental model was proposed [70,71] accounting for the iron concentration in blood, CSF, and brain extracellular fluid (Interstitial Fluid, ISF). Its parameters estimate the iron intake from blood directly to ISF or mediated by the passage to the CSF and from the CSF to ISF, and, obviously, the iron clearance into the blood or again mediated by the CSF (see arrows in Figure 1) [70,71,72,73].

In Figure 1, the scheme of the compartmental model is reported. The main variables are:Iron concentration in blood (mg/L)—red compartment;Iron concentration in CSF (mg/L)—blue compartment;Iron concentration in ISF (mg/L)—yellow compartment;

While the most important parameters are:Blood → ISF: Kinetic constant rate for iron entering from blood to brain (consequently ISF), across BBB;Blood → CSF: Kinetic constant rate for iron entering from blood to CSF across the blood–CSF barrier (BCSFB);CSF → ISF: Kinetic constant rate for iron passing from CSF to ISF;ISF → Blood: Kinetic constant rate for iron returning from the brain to the blood.CSF → Blood: Kinetic constant rate for iron returning from CSF and brain to blood;ISF → CSF: Kinetic constant rate for iron passing from ISF to CSF;

A short description of the underlying mechanisms is here proposed. For further details and an explanation of the whole model, refer to [70,71].

#### 2.3.1. Blood → ISF

The brain shares the same iron regulatory mechanisms as the systemic organs, focused on the maintenance of iron homeostasis [74]. The uptake of iron is started by the transferrin–receptor pathway in the brain endothelium through brain microvascular endothelial cells [75].

Iron transport into the brain occurs at the BBB, which is composed of microvascular endothelial cells and is supported by astrocytes, neurons, and microglia. It is a very selective border that can be trespassed by iron either by a non-heme or heme transporter.

As concerns non-heme transporters, iron uptake occurs through either transferrin-bound iron (TBI) or non-transferrin-bound iron (NTBI) mechanisms. The TBI uptake pathway involves a series of steps, including the binding of iron to the Tf protein, the binding of Tf to the TfR, their endocytosis, and the iron reduction and release into the cytosol through the divalent metal transporter 1 (DMT1) [76]. Among the iron transport systems, the key players are ferroportin (FPN), the transporter mediating iron efflux from cells; DMT1, ZIP8 and ZIP14, which, on the contrary, mediate iron influx into the cytoplasm, acting on the plasma or lysosomes and endosome membranes; and mitoferrin, involved in iron transport into the mitochondria for heme synthesis and Fe-S cluster assembly [77]. TBI uptake seems to be slightly affected by AD [78], while NTBI has been proven to be responsible for iron overload in the mouse brain, especially in cases of acute inflammation [79].

Another non-heme transporter is discussed in [4]: the transfer of iron by *Porphyromonas gingivalis* Outer Membrane Vesicles (OMVs) from the bloodstream to the brain may provide one explanation for the iron maldistribution observed in AD. Of particular note, the proliferation of *P. gingivalis* and the disruption of the epithelia during periodontitis progression enable more OMVs to spread into circulation and cause a shift in iron distribution from blood to brain, suggesting that OMVs may serve as a mechanistic link between chronic periodontitis and AD [4]. OMVs are spherical nanostructures released from the outer membrane of Gram-negative bacteria. Carrying a broad range of bacterial virulence factors, they have a pivotal role in bacterial growth, intercellular communication, biofilm formation, invasion, and modulation of host defense [80].

Heme transporters are, instead, due to microhemorrhages and BBB focal damages. Similar to traumatic brain injury, they can cause small quantities of hemoglobin to trespass the BBB to reach the ISF. Here, hemoglobin degradation in carbon monoxide (CO), iron (Fe^2^⁺), and biliverdin, which is rapidly converted to bilirubin (BR), may occur catalyzed by Heme oxygenase 1 (HO-1) [81,82,83]. HO-1 is up-regulated in the brains of people with AD and PD, and HO-1 induction in primary astroglia cultures promotes the deposition of non-transferrin iron, mitochondrial damage, and macroautophagy [81,82]. It also predisposes co-cultured neuronal elements to oxidative injury, and HMOX1 transgenic mice selectively over-express human HO-1 in the astrocytic compartment, exhibiting at 48 weeks increased deposits of glial iron in the hippocampus and other subcortical regions without overt changes in iron-regulatory and iron-binding proteins relative to age-matched wild-type [82,83,84].

Finally, free heme is scavenged by hemopexin, but in AD patients, the binding between hemopexin and heme may be disrupted, allowing free heme to interact with amyloid β, further promoting Tau aggregation and oxidative stress [85].

#### 2.3.2. Blood → CSF and CSF → ISF

The cells of the choroid plexus extend a net throughout all four cerebral ventricles, elaborating the CSF, and receive a high blood supply, which is filtered by BCSFB. This membrane, although far less selective in comparison with the BBB, plays an important role in uptaking many nutrients, such as iron. On the contrary, there are no diffusional barriers between the CSF and the brain interstitial fluid ISF, and iron can be bound by the transferrin synthesized by oligodendrocytes and circulate throughout the brain interstitium, supplying iron to the CNS cells [86].

#### 2.3.3. ISF → Blood

Ferrous iron efflux is supported by FPN and requires the subsequent ferrous iron oxidation—ferroxidation—by a ferroxidase, either ceruloplasmin (CP) or hephaestin (HP). Normally, this efflux is regulated by the abundance of FPN in the plasma membrane, where the peptide hormone, hepcidin, binds to FPN and triggers its internalization and degradation [76,87]. It is reported that a sequence within the E2 domain of soluble amyloid precursor protein (sAPP) binds to ferroportin in the basal membrane of human brain microvascular endothelial cells of the BBB, stabilizing ferroportin in the membrane and stimulating iron efflux at the brain side of these cells [88].

#### 2.3.4. ISF → CSF and CSF → Blood

The clearance of iron, controlled by bulk CSF flow and/or by the removal mechanism in the BCSFB, back to the blood circulation is of capital importance.

Calculations of the amount of iron leaving the system with the bulk flow of CSF indicate that most iron entering the brain across the capillary endothelium finally leaves the system with the bulk outflow of CSF through arachnoid villi and other channels [89]. A system in which the influx of iron into the brain is regulated by receptor-mediated transport and in which efflux is by bulk flow is ideal for the homeostasis of brain iron [89].

Indeed, a modification in this exchange could lead to an accumulation or a lack of iron in the brain, and both scenarios can become pathological.

The CSF acts as a sort of dustbin collecting all the waste from the ISF, and is the fluid where Aβ and Tau protein samples are collected for AD diagnosis [86].

Also, the role of the ‘glymphatic’ system, a network of perivascular pathways supporting exchanges between CSF and ISF and contributing to the efflux of interstitial solutes, is of great importance [90].

Finally, we can conclude that several mechanisms are responsible for the iron balance in the brain intracellular fluid and ISF, making it difficult to understand which one is preponderant and should be regulated to avoid brain iron overload. The possibility of a direct, local intervention on the ISF is therefore advised.

### 2.4. Towards Novel Intrathecal Therapy: The Nano-Chelating Approach

As far as AD therapies are concerned, 53,484 papers were found on WOS in July 2023 using “Alzheimer’s Disease” AND “therapy” as keywords. However, a resolutive and effective management reducing the values of the main biomarkers in any disease phase is still missing.

Controlling iron’s level is not the only innovative route for therapies. The Food and Drug Administration (FDA) recently approved monoclonal antibodies Aduhelm (aducanumab) and Lecanemab, respectively, which target soluble oligomers clearing amyloid plaques and contributing to slowing AD progression [91].

Side effects, such as microhemorrhages and brain swelling, highlight the need for improved therapy and novel diagnostic tools for assessing efficacy [92].

Other approaches target the defective adenosine-triphosphate-(ATP)-binding cassette (ABC) transporter implicated with AD progression and are under both in vitro and in vivo preclinical studies [93,94].

Iron chelation has been introduced as a new therapeutic strategy for the treatment of neurodegenerative diseases that have a component of metal ion accumulation.

Iron chelators have been previously used to remove the excess of toxic iron in patients with systemic iron overload. Chelating agents form iron complexes that promote iron excretion, clear plasma non-transferrin-bound iron, remove excess iron from cells, and restore iron levels in the body [95]. Furthermore, natural chelators, such as quercetin and curcumin, proved effective and not toxic [96].

However, many of them, such as Deferoxamine (DFO), have a considerably limited short half-life, non-specific tissue distribution, and off-target toxicities and need an extremely precise dose regimen with daily infusion [97]. DFO has a linear structure and can wrap around iron, chelating it in all six coordination sites and ‘isolating’ the iron ion from the outside.

At first, DFO was approved by the FDA as an iron chelator drug. Then, deferiprone and deferasinox were introduced as iron-chelating drugs. The pharmacological and toxicological aspects of these three drugs have been widely investigated [98].

The safety and effectiveness of deferiprone in patients with Prodromal Alzheimer’s Disease (pAD) and Mild Alzheimer’s Disease were investigated. In particular, a clinical trial [99] was carried out to evaluate the capability of the drug to slow the cognitive decline in AD patients orally administered at high doses. Indeed, one of the drawbacks of iron-chelating drugs is the high dosage required to achieve the maximum efficacy, which can affect the pharmacokinetics, metabolic and toxicological parameters. Another limitation of this therapeutic approach is its poor selectivity. Therefore, the risk/benefit assessment must be carefully considered in protocols. In this context, to overcome the drawbacks and improve its efficacy, different iron-chelator delivery approaches have been designed and investigated, such as polymer conjugation or encapsulation in nanoparticulate systems [100,101].

Indeed, as an instrument of chelating therapy, nanoparticles can be used, which can be prepared in different ways [102] to exploit their functions and properties. Iron-chelating-loaded/decorated nanocarriers might be functionalized to target specific tissues and might release the iron-chelating drugs with slow kinetics, allowing a dose decrease with a limited number of side effects.

Chelating nanoparticles capable of crossing the BBB via receptor-mediated systems have been exploited [103] and shown to be able to reduce the iron ions toxicity [104], but some concern about the possible sequestration of iron from storage proteins as ferritin in brain tissues has been expressed [105]. Interestingly, the effectiveness of this strategy was demonstrated in a mouse model of AD [106]. However, the real advantages are still controversial [107]. Indeed, to the best of our knowledge, no experiments on humans using iron-chelator nanoparticles are present, nor has long-term toxicity been already studied [108]. Most of the studies are on human cell lines or, at most, on animal models.

Anyway, the toxicology and longitudinal studies will also be considered for the nanodelivery systems.

Recently, the design and development of novel multifunctional iron chelator polymer-shelled nanobubbles as a potential therapeutic tool for preventing neurodegenerative disease, controlling Fe^3+^ concentration, and reducing ROS concentration in the brain has been pursued as a new nanotechnology application, in addition to other approaches [109].

The tunable polymer-shelled nanobubbles are vesicle-like nanostructures comprising a perfluorocarbon core stabilized by a Langmuir monolayer able to strongly interact with polysaccharide chains and suitable for drug delivery across very selective membranes [110,111,112,113].

The same research group synthesized GC-DFO conjugate nanobubbles (GC-DFO NBs), able to effectively chelate Fe^3+^, forming a complex able to retain or even enhance the chelating ability of DFO at low iron concentrations [114].

By detecting with Thioflavin T fluorescence the presence of amyloid aggregates (Aβ1-42), a high signal was observed in the presence of the iron chelators (DFO, GC-DFO, and NB-GC-DFO) in comparison with Fe^3+^, confirming that iron chelation can effectively prevent Aβ misfolding [114].

Many safety issues were addressed: GC-DFO NBs showed no hemolytic activity on red blood cells and cytocompatibility with different cell lines. Also, NBs were tested on organotypic brain culture cells *(Substantia nigra*) before and after iron chelation.

No sign of NBs toxicity was present neither in the MTT nor in the LDH test at proper dilution, and no induced DOPAn loss was detected [114].

Administration of substances directly into the CSF and the intrathecal space that surrounds the brain and spinal cord is one approach that can circumvent the BBB to enable drug delivery to the CNS. Interestingly, implantable systems can allow both the injection and the collection of samples of CSF to monitor the therapeutic effectiveness [115,116].

However, previous ‘in vivo’ experiments [117] and models [118] showed that intrathecally injected nanoparticles in the CNS mainly concentrate in the spinal cord, according to the fact that the BCSFB promotes the clearance of brain waste but not the entrance.

Therefore, the application of a physical external stimulus to reach the CNS bypassing biological barriers is needed. For systemic administration, Focused Ultrasound (FUS) can open the BBB. For intrathecal administration, the application of an external magnetic field able to drive magnetic nanosystems to the desired location can be investigated [119,120].

For this purpose, GC-DFO NBs can be decorated with cleavable Superparamagnetic Iron Oxide Nanoparticles (SPIONs), (synthetic γ-Fe_2_O_3_ (maghemite) or Fe_3_O_4_ (magnetite) particles ranging between 10 nm and 100 nm in diameter), which are useful as a promising tool in magnetic nanosystems for AD theranostics [121], becoming GC-DFO_MNBs. Previous research showed the capability to combine SPIONs with polymer-shelled nanobubbles [122].

Recently, the advances in and problems with the use of magnetically guided and magnetically responsive nanoparticles in drug delivery and magnetofection were reviewed [123]. Non-invasive magnetically guided delivery of magneto-electric nanocarriers was developed, proving that the nanocarriers are uniformly distributed inside the brain and are non-toxic to the brain and other major organs [124].

Interestingly, a framework for building optimal delivery systems that uses nanoparticle–biological interaction data and computational analyses can be used to guide future nanomaterial designs and delivery strategies [125]. Recent evidence has focused on driving nanoinformatics research to develop innovative and integrated tools for in silico nanosafety assessment [126] and using machine learning tools to predict the functions of nanoparticles [127,128].

Moreover, mathematical models for the dynamics of superparamagnetic nanoparticles under the influence of applied magnetic fields were studied for many applications, e.g., magnetic drug targeting in cancer therapy [129].

## 3. Conclusions and Future Perspectives

Iron accumulation is a critical point in neurodegenerative diseases. A therapeutic and personalized approach aiming at nanochelating the iron overload in the brain fluid using an intrathecal-delivering implant seems feasible based on both in silico and in vitro preliminary investigations.

Provided the nanochelating agents are effective and safe, their dosage can be tuned according to the samples of the CSF extracted by the same intrathecal implant, allowing a real-time assessment of the waste from the brain, including iron. Mathematical modeling can provide a useful approach to investigating in silico the modulation and effects of novel nanochelating agents.

## Figures and Tables

**Figure 1 ijms-25-02337-f001:**
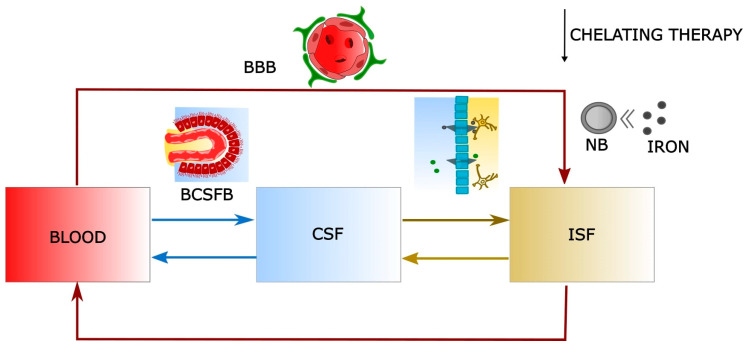
Schematic representation of a model macroscopically describing iron flux and potential therapeutic chelating strategy acting directly into the brain environment. BBB takes into account the entry of iron from the blood directly into the brain. Iron can also enter from blood to CSF across the Blood–CSF Barrier and arrive at the ISF. Iron can pass from ISF to CSF and return from CSF and brain to blood. Chelating therapy could be based on nanochelating agents able to chelate iron, with the help of mathematical modeling to tune and monitor the chelating action.

**Table 1 ijms-25-02337-t001:** Direct and indirect iron-related damaging mechanisms. For each direct mechanism (A1, A2, A3, A4), corresponding potential indirect mechanisms (B1, B2, B3, B4) are shown (A1 → B1; A2 → B2; A3 → B3; A4 → B4).

A. Direct Mechanisms	B. Indirect Mechanisms
A1. oxidative stress by Fenton and Haber–Weiss reaction	B1. iron-induced oxidation affects the lipid metabolism regulated by the apolipoprotein
A2. direct binding to amyloidogenic proteins	B2. altered iron homeostasis leads to mitochondrial dysfunction
A3. alteration of the spontaneous neuronal activity	B3. iron-calcium interplay modulates neuronal functionality
A4. enhancement of microbial proliferation	B4. iron-fueled infections enhance amyloid plaques formation

## Data Availability

No new data were created.

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
