# Peer review of "Iron Overload in Brain: Transport Mismatches, Microbleeding Events, and How Nanochelating Therapies May Counteract Their Effects"

_ijms, 2024, doi:10.3390/ijms25042337_

Round 1

Reviewer 1 Report

Comments and Suggestions for Authors

This comprehensive review article summarizes iron overload in Alzheimer's disease pathogenesis and explores potential therapies involving iron chelators. While the manuscript is well-written and potentially interesting, I suggest a few improvements before publication.

1)              In Section 2.1, the authors mentioned direct iron measurement methods; however, the sentence lacks sufficient explanation, making it challenging to understand why these methods are considered unreliable. I believe a brief description is needed.

2)             In Section 2.2.1, the authors described the ferroptosis mechanism. However, the sentence lacks sufficient detail to explain ferroptosis adequately. Ferroptosis is a crucial mechanism associated with iron overload. This section requires an explanation of ferroptosis mechanisms, including lipid oxidation, transferrin function, ferritinophagy, and other relevant factors.

3)              In Section 2.2.1, the author described several AD model mice, including the 'PSAPP mouse model,' 'APP/PS1 transgenic mice,' and 'AD mice.' It is unclear whether these models are on the same line or different lines. If the latter is the case, the authors need to provide a brief description of the features of these AD models and their iron-related abnormalities.

4)             In the table, the authors summarized both direct and indirect iron-related mechanisms. It might enhance comprehension if these mechanisms were accompanied by diagrams for better understanding.

5)              In Section 2.2.2, the authors listed genes involved in iron abnormalities in AD pathology. However, there are no descriptions of iron involvement in the sentence describing 'APOE-e4.' If this gene is not related to iron abnormalities, consider moving this sentence elsewhere, as APOE-e4 is known to be significant in AD pathology.

6)             In Section 2.3.1, the sentence describing iron transport is a bit challenging to understand. It would be helpful to reorganize the information to clearly illustrate how iron transport works in normal states and then compare it with AD pathology.

7)              In Section 2.4, the authors described recent treatments, including antibody medicines and ABC transporters, for AD. However, it is unclear if there is any relation between iron and these targets. If not, these sentences may not correspond well to the title. Consider rephrasing for better understanding.

I hope that my comment is useful for the improvement of the article. 

Reviewer 2 Report

Comments and Suggestions for Authors

1.      The article provides an overview of the mechanisms leading to iron accumulation in the brain. However, a more detailed exploration of the underlying molecular and cellular mechanisms, including the role of genetic factors, environmental influences, and their interactions, could provide a deeper understanding of the problem and inform more targeted interventions.

2.      While the article mentions that novel formulations of nanovectors have been tested in organotypic brain models, the transition from in silico and in vitro models to in vivo human models is a significant step. The article could address the current state of research regarding the efficacy of these therapies in human models, including any clinical trials, their outcomes, and the challenges faced.

3.      The article discusses the potential of nanochelating therapies to counteract the effects of iron overload in the brain. However, the specificity of these therapies to target only excess iron without affecting the essential iron required for normal physiological functions might be a concern. The article could benefit from a more detailed discussion on the selectivity of these therapies and their safety profiles

4.      The article discusses the potential of nanochelating therapies as a novel approach to treating neurodegenerative diseases associated with iron overload. However, the long-term outcomes, sustainability, and potential side effects of these therapies remain unclear. A discussion on these aspects, including any available longitudinal data, would strengthen the article's conclusions.

5.      The article could benefit from a comparative analysis of nanochelating therapies with existing treatments for neurodegenerative diseases associated with iron overload. This analysis could include efficacy, safety, cost-effectiveness, and patient quality of life, providing a more comprehensive picture of where nanochelating therapies stand in the current treatment landscape. Also briefly discuss how the described iron transport mechanisms are linked to neurodegenerative diseases, especially Alzheimer’s. This connection can help readers understand the relevance of your model in the context of disease pathology.

6.      In line 123, the manuscript should provide additional elucidation on the significance and functional roles of furin and brain-derived neurotrophic factor (BDNF) within the given context.

7.      Include additional information on the nature and mechanism of the allosteric NEAT dominion, elucidating its relationship with iron.

8.      Discuss the potential integration of different imaging techniques, such as MRI and PET in a more connected manner to offer a comprehensive view of early detection methods. Expand on the challenges of current iron detection methods in CSF and discuss potential ways these could be improved or alternatives explored.

9.      Provide a bit more clarification on the iron clearance pathways, possibly summarizing key findings related to the clearance mechanisms, emphasizing their significance in maintaining iron homeostasis.

10.  Please include the full name of the BCSFB acronym in Figure's 1 legend.

11.  In section “2.3.4 ISF → CSF and CSF → Blood”, please elaborate on the “capital importance” mentioned.

12.  In section “2.4 Towards novel intrathecal therapy: the nanochelating approach”. Although this section does mention some limitations of certain chelating agents, such as DFO, I believe mentioning the limitations of nanoparticles themselves is important. This is mentioned in various studies including this review article by Dr. Elmarguzi et al. (limitations on page 4 of pdf linked below, blood- brain barrier and Alzheimer’s page 11), which could benefit you in other aspects such as including additional details about the consumption and types of nanoparticles.

Dr. Elmarguzi’s article: https://doi.org/10.59049/2790-0231.1244

PDF https://pmpj.najah.edu/cgi/viewcontent.cgi?article=1244&context=journal

13.  Spelling mistake in line 220, replace “lavel” with level

Round 2

Reviewer 2 Report

Comments and Suggestions for Authors

Thank you for taking all comments into consideration; the manuscript is better now.